# Probabilistic Circuits for Variational Inference in Discrete Graphical Models

**Andy Shih**
Computer Science Department
Stanford University
andyshih@cs.stanford.edu

**Stefano Ermon**
Computer Science Department
Stanford University
ermon@cs.stanford.edu

## Abstract

Inference in discrete graphical models with variational methods is difficult because of the inability to re-parameterize gradients of the Evidence Lower Bound (ELBO). Many sampling-based methods have been proposed for estimating these gradients, but they suffer from high bias or variance. In this paper, we propose a new approach that leverages the tractability of probabilistic circuit models, such as Sum Product Networks (SPN), to compute ELBO gradients exactly (without sampling) for a certain class of densities. In particular, we show that selective-SPNs are suitable as an expressive variational distribution, and prove that when the log-density of the target model is a polynomial the corresponding ELBO can be computed analytically. To scale to graphical models with thousands of variables, we develop an efficient and effective construction of selective-SPNs with size $O(kn)$, where $n$ is the number of variables and $k$ is an adjustable hyperparameter. We demonstrate our approach on three types of graphical models – Ising models, Latent Dirichlet Allocation, and factor graphs from the UAI Inference Competition. Selective-SPNs give a better lower bound than mean-field and structured mean-field, and is competitive with approximations that do not provide a lower bound, such as Loopy Belief Propagation and Tree-Reweighted Belief Propagation. Our results show that probabilistic circuits are promising tools for variational inference in discrete graphical models as they combine tractability and expressivity.

## 1 Introduction

Variational methods for inference have seen a rise in popularity due to advancements in Monte Carlo methods for gradient estimation and optimization [11]. The advent of black box variational inference [35] and the re-parameterization trick [18, 36, 39] have enabled the training of complex models with automatic differentiation tools [21], and provided a low-variance gradient estimate of the ELBO for continuous variables.

We are interested in extending the success of variational methods to inference in discrete graphical models. Discrete graphical models are used in numerous applications, including physics, statistics, and machine learning tasks such as topic modelling and image segmentation. One of the central problems of probabilistic inference in these graphical models is computation of the log partition function [14, 20]. Exploring richer variational families beyond fully factored mean-field would enable a tighter bound of the log partition function, and in turn bound the probability of evidence.

Optimization of expressive variational families in discrete settings, however, is difficult because the re-parameterization trick for computing gradients does not extend naturally to discrete variables. Roughly speaking, the re-parameterization trick pushes away the randomness of sampling inputs, and instead applies a differentiable transformation onto a base distribution containing the randomness. Unfortunately, this cannot easily be extended to discrete variables because there is no differentiable

transformation that maps to a discrete distribution. As such, a number of techniques have been proposed to make Monte Carlo gradient estimates work with discrete models. These techniques range from score function estimation [45, 27] to continuous relaxations [26, 13, 41, 42], but they still suffer from high variance. Optimizing the ELBO for inference tasks using noisy gradients with high variance is impractical, especially in high dimensions.

Instead, we revisit the choice of Monte Carlo methods for evaluation and optimization of the ELBO of discrete models. We examine the alternative approach of computing the ELBO analytically, avoiding the high variance of Monte Carlo methods. Whereas Monte Carlo methods use a few representative samples for an efficient, unbiased, but high variance estimate, we exhaustively consider all possible inputs to get an exact gradient computation with zero bias or variance. At first glance, such exact computation of the ELBO appears intractable, and seems to limit ourselves to variational distributions such as mean-field or possibly structured mean-field [14, 37]. Fortunately, we can use tools from probabilistic circuits to bridge the gap between expressivity and tractability.

Probabilistic circuits have been used for state-of-the-art discrete density estimation [9, 24]. They are fully expressive – they can express any density over discrete variables – and gain their tractability properties by enforcing global constraints in the circuit structure, which come at the cost of succinctness [7, 3]. Varying degrees of structural constraints give rise to different families of probabilistic circuits, and can allow for polynomial time computation of marginals, moments, and more [34, 16]. For example, the constraint of decomposability alone leads to Sum-Product-Networks (SPNs), while the additional constraints of structured decomposability and prime partitioning lead to Probabilistic Sentential Decision Diagrams (PSDDs) [34, 19, 5].

In this paper, we propose the use of probabilistic circuits as an expressive variational family for inference in discrete graphical models. We prove that for graphical models with polynomial log-density, the class of selective and decomposable circuits enables exact computation of the corresponding ELBO. In particular, we provide an algorithm to compute the ELBO in time linear in the size of the probabilistic circuit, which can then be combined with automatic differentiation to give exact gradients of the ELBO for optimization.

Selective and decomposable probabilistic circuits, otherwise known as selective-SPNs [31], are more succinct than circuit families such as PSDDs. PSDDs have been shown to scale to various density estimation and classification tasks [24, 4], and thus selective-SPNs can only scale better. For even greater scalability and ease of use, we present an effective linear time construction of selective-SPNs with size $O(kn)$, where $n$ is the number of variables and and $k$ is an adjustable hyperparameter. Our construction enables us to scale to graphical models with thousands of variables and terms/factors.

Our contributions are as follows:

- For graphical models where the log density is a polynomial, we show that selective-SPNs can serve as an expressive variational distribution that enables exact ELBO computation. We prove that the corresponding ELBO can be computed in $O(tm)$ time, where $t$ is the number of terms in the polynomial and $m$ is the size of the probabilistic circuit.

- We present an algorithm for constructing selective-SPNs of size $O(kn)$, where $n$ is the number of variables and $k$ is an adjustable hyperparameter. Constructing selective-SPNs with linear dependency on $n$ allows us to scale to graphical models with thousands of variables and terms/factors. Automating this construction enables easy integration of our framework into new inference tasks.

- We validate our approach on three sets of discrete graphical models: Ising models, Latent Dirichlet Allocation, and factor graphs from the UAI Inference Competition. We show that variational inference with selective-SPN gives better lower bounds of the log partition function than mean-field and structured mean-field variational inference. In addition, our approach is competitive with other approximation techniques that do not provide a lower bound, such as Loopy Belief Propagation and Tree-Reweighted Belief Propagation.

## 2 Preliminaries

### 2.1 Variational Inference

For many probabilistic models, we have access to unnormalized probability densities $w(\mathbf{x})$ but we cannot easily compute the partition function, or marginals in general [23, 20]. For discrete models, the probability density and partition function can be written as follows:

$$p(\mathbf{x}) = \frac{w(\mathbf{x})}{Z} \quad , \quad Z = \sum_{\mathbf{x} \in \mathbf{X}} w(\mathbf{x})$$

There are many ways to estimate the partition function, and one such approach is variational inference, which poses approximate inference as an optimization problem [44, 2]. Variational inference works by first choosing a variational family of distributions $q_\phi$ with tractable density evaluation and that we can easily sample from. Then we can rewrite $Z$ with importance weights. Finally, to work with log probabilities, we take the log of both sides to get the Evidence Lower Bound (ELBO), where $H(.)$ denotes the entropy of a distribution.

$$Z = \mathbb{E}_{\mathbf{x} \sim q_\phi} \left[ \frac{w(\mathbf{x})}{q_\phi(\mathbf{x})} \right] \quad , \quad \log Z \geq \mathbb{E}_{\mathbf{x} \sim q_\phi} \left[ \log w(\mathbf{x}) \right] + H(q_\phi)$$

To get the best estimate of $\log Z$, we maximize the ELBO by gradient descent with respect to parameters $\phi$. The optimization can be done using black-box variational inference [35, 21] by computing a noisy gradient from samples of the variational distribution using automatic differentiation tools. If $\mathbf{x}$ is continuous and $q_\phi$ is re-parameterizable [18], we can simply sample $\mathbf{x} \sim q_\phi$, evaluate the ELBO, and backpropagate using the re-parameterization trick. However, for discrete $\mathbf{x}$ we cannot backpropagate through samples unless we resort to continuous relaxations [26, 13, 41, 42] or use REINFORCE-style estimations [45, 27], which introduce bias or variance.

We revisit the choice of Monte Carlo evaluation: can we obtain gradients without backpropagating through samples? That is, can we avoid sampling altogether, and pick a variational family of distributions that enables us to compute the ELBO analytically while maintaining full expressiveness? If so, we can backpropagate on the exact ELBO value to obtain its gradients with respect to $\phi$, with zero bias and zero variance. Simple variational families such as mean-field often allow us to compute the ELBO exactly, but they sacrifice expressiveness. To support analytic computation of the ELBO and maintain full expressiveness, we look towards probabilistic circuit models.

### 2.2 Probabilistic Circuits

Probabilistic circuits are directed acyclic graphs (DAGs) over input variables $\mathbf{X}$ containing *sum*, *product*, and *leaf* nodes [6, 34]. Each child edge of a sum node has a corresponding weight, and each leaf node has a distribution over an input variable. For simplicity we assume the leaf nodes are univariate, but they can be multivariate in general [40]. An example is shown in Figure 1.

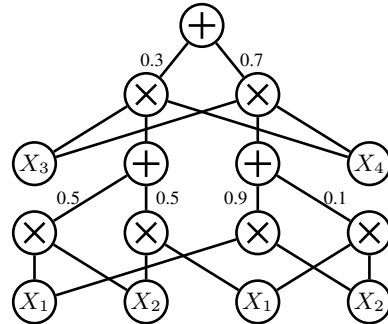

Probabilistic circuits can be viewed as computation graphs that feed inputs into the leaf nodes, take weighted sums at sum nodes and take products at product nodes. The value obtained at the root node of the graph is treated as the output of the circuit. More formally, for a leaf node $i$ we let $l_i$ denote its leaf distribution. For an internal node $i$ we let $ch(i)$ denote the set of its children. For a sum node $i$ and one of its children $j$, we let $a_{ij}$ denote the edge weight from node $i$ to $j$. Then, the distribution $g_i(\mathbf{x})$ of node $i$ on an input $\mathbf{x}$ can be expressed as follows:

Figure 1: A decomposable probabilistic circuit over 4 variables. If $X_1$ leaf nodes are disjoint and $X_2$ leaf nodes are disjoint, then the circuit is also selective.

$$g_i(\mathbf{x}) = \begin{cases} l_i(\mathbf{x}) & i \text{ is leaf node} \\ \sum_{j \in ch(i)} a_{ij} g_j(\mathbf{x}) & i \text{ is sum node} \\ \prod_{j \in ch(i)} g_j(\mathbf{x}) & i \text{ is product node} \end{cases}$$

When the leaf nodes and the edge weights are positive, the output of the circuits are also positive, so they can be used to encode a probability density. Probabilistic circuits are useful for modeling densities because they support efficient sampling and can compute exact likelihoods, marginals, and more when they satisfy some of the following properties [8].

**Definition 1.** *Let $vars_i$ denote the set of variables that appear at or below node $i$. A probabilistic circuit is **decomposable** if for every product node $d$, the variable sets of its children are pairwise disjoint: for all $i, j \in ch(d)$, with $i \neq j$, we have that $vars_i \cap vars_j = \emptyset$.*

**Definition 2.** *A probabilistic circuit is **selective / deterministic** if under any input $\mathbf{x}$, at most one child of each sum node evaluates to non-zero.*

Furthermore, there are stronger properties such as structured decomposability, prime-partitioning, that enable more advanced operations to be computed in polynomial time on the size of the circuit [19, 38]. Different subsets of these properties lead to different families of probabilistic circuits, such as PSDDs, SPNs [34, 19], and selective-SPNs [31].

Since the internal nodes are sums and products, probabilistic circuits can naturally be seen as a compact representation of a polynomial over the leaf nodes. Common choices of leaf distributions are Gaussians for continuous variables, and categorical for discrete variables [34]. It is then easy to see that these circuits are fully expressive and can approximate any density, by viewing them as hierarchical mixture models [32, 43].

The size of a probabilistic circuit refers to the number of edges in the circuit. In the rest of this paper, we will assume probabilistic circuits of size $m \geq n$, the number of variables, and that the probabilistic circuits are smooth (i.e. children of each sum node cover the same set of variables [7]). We also assume that edge weights under any sum node adds up to 1, since this is enforceable with $O(m)$ operations [32]. Lastly, we assume the discrete variables are binary, taking on values in either $\{-1, 1\}$ or $\{0, 1\}$, but our framework can be easily extended to general discrete settings by using discrete leaf distributions in the probabilistic circuit.

# 3 Probabilistic Circuits as the Variational Distribution

In this section, we explore the use of probabilistic circuits as a variational distribution – we want to optimize the model parameters to most closely match a target distribution. Although probabilistic circuits have strong tractability properties for sampling and evaluation, it is unclear how to compute the exact ELBO for arbitrary choices of density $w$. For example, when the probabilistic circuit encodes a uniform distribution, computing the ELBO is as hard as summing out $\log w(\mathbf{x})$, which can be designed to be #P-hard if there are no restrictions on $w$ [17]. However, we show that for models whose log density can be written as a polynomial, the corresponding ELBO with respect to a selective-SPN can be computed analytically.

## 3.1 Exact Computation of ELBO

Exponential-family models where the log density is a polynomial commonly arise when working with graphical models [20, 12]. This refers to unnormalized probability densities written as:

$$v(\mathbf{x}) = \sum_{f \in F} f(\mathbf{x}) \quad , \quad w(\mathbf{x}) = e^{v(\mathbf{x})}$$

where each factor $f$ is a monomial (a single-term polynomial). For discrete models, this form can actually express any density, but with possibly exponentially many factors, by considering the Discrete Fourier Transform [30].

**Lemma 1.** *Every function $v : \{-1, 1\}^n \to \mathbb{R}$ can be expressed as a polynomial [30].*

The same can be shown for variables taking values in $\{0, 1\}$. In practice, many factor graphs such as Ising models have unnormalized probability densities that can be written as a polynomial with few terms. We would like to bound the partition function of these models using variational inference. This involves the following ELBO computation.

$$\log Z \geq \mathbb{E}_{\mathbf{x} \sim q_\phi} \Big[ \sum_{f \in F} f(\mathbf{x}) \Big] + H(q_\phi) = \sum_{f \in F} \mathbb{E}_{\mathbf{x} \sim q_\phi} \Big[ f(\mathbf{x}) \Big] + H(q_\phi)$$

In other words, we need to compute the expectation of each monomial with respect to our variational distribution $q_\phi$, and compute the entropy $H(q_\phi)$. In general these are not easy, but we can leverage some of the properties of probabilistic circuits to enable this computation.

**Theorem 1.** *Given a discrete model $p$ whose log density can be written as a polynomial, and a variational distribution $q_\phi$ that is a selective-SPN, the ELBO of the partition function of $p$ can be computed in time $O(tm)$, where $t$ is the number of terms in the polynomial and $m$ is the size of $q_\phi$.*

*Proof.* First, we show that the expectation of each monomial $f$ with respect to $q_\phi$ can be computed in $O(m)$. We write $x \in f$ if variable $x$ appears in the monomial $f$, and assume $f$ has no coefficient for the moment. We consider the 3 types of nodes, and let $g_i$ be the distribution induced by node $i$.

At a sum node $i$ (left), we have linearity of expectations. At a product node $i$ (right), since children are decomposable and have disjoint variable sets, they are independent and the expectation decomposes.

$$\mathbb{E}_{\mathbf{x}\sim g_i}\Big[f(\mathbf{x})\Big] = \sum_{j\in ch(i)} a_{ij}\mathbb{E}_{\mathbf{x}\sim g_j}\Big[f(\mathbf{x})\Big] \qquad \mathbb{E}_{\mathbf{x}\sim g_i}\Big[f(\mathbf{x})\Big] = \prod_{j\in ch(i)} \mathbb{E}_{\mathbf{x}\sim g_j}\Big[f(\mathbf{x})\Big]$$

Finally, at a leaf node $i$ with univariate distribution over variable $x$ with mean $\mu$:

$$\mathbb{E}_{x\sim g_i}\Big[f(x)\Big] = \begin{cases} \mu & \text{if } x \in f \\ 1 & \text{otherwise} \end{cases}$$

The base case does not evaluate $f$ directly because we ignored the coefficient of the monomial $f$. As a final step, we multiply the result by the original coefficient of $f$. Since computation of each node takes linear time in the number of children, the expectation of a monomial takes $O(m)$ operations, leading to $O(tm)$ operations in total. Next we show that the entropy term can be computed in $O(m)$. At a sum node, we have the following due to selectivity:

$$-\mathbb{E}_{\mathbf{x}\sim g_i}\Big[\log g_i(\mathbf{x})\Big] = \quad -\sum_{j\in ch(i)} \alpha_{ij}\mathbb{E}_{\mathbf{x}\sim g_j}\Big[\log \sum_{k\in ch(i)} \alpha_{ik}g_k(\mathbf{x})\Big]$$

$$= -\sum_{j\in ch(i)} \alpha_{ij}\mathbb{E}_{\mathbf{x}\sim g_j}\Big[\log \alpha_{ij}g_j(\mathbf{x})\Big] = \quad -\sum_{j\in ch(i)} \alpha_{ij}\log\alpha_{ij} + \alpha_{ij}\mathbb{E}_{\mathbf{x}\sim g_j}\Big[\log g_j(\mathbf{x})\Big]$$

Because of selectivity, at most one child can evaluate an input to non-zero. Therefore, the expectation of an input sampled from the distribution of one child with respect to another child must be zero, hence we can simplify the summation inside the expectation.

At a product node, decomposability again implies independence between children, so the entropy is the sum of the entropy of the children.

$$-\mathbb{E}_{\mathbf{x}\sim g_i}\Big[\log g_i(\mathbf{x})\Big] = -\sum_{j\in ch(i)} \mathbb{E}_{\mathbf{x}\sim g_j}\Big[\log g_j(\mathbf{x})\Big]$$

Lastly, we can compute the entropy of the Bernoulli leaf nodes in constant time. $\qquad\square$

We have shown that we can tractably compute the corresponding ELBO (and therefore its gradients via automatic differentiation) for selective-SPNs and polynomial log-densities. Additionally, the computation of the entropy of selective-SPNs in linear time may be of independent interest, for tasks beyond the computation of the ELBO.

## 4  Efficient Construction of Selective-SPNs

In the interest of scalability and ease of use, we now describe an algorithm for constructing selective-SPNs of size linear in the number of variables. Our algorithm generates selective and decomposable probabilistic circuits that can then be plugged in as the variational distribution for discrete graphical models, as shown in Section 3. Essentially, we are constructing the structure of the selective-SPN without considering the target model, in a similar spirit to RAT-SPNs [33]. These selective-SPNs are more expressive than mean-field or structured mean-field variational families, which will be reflected in the experimental results in Section 5.

Our algorithm takes as input the number of variables $n$ and an integer $k$ denoting the size budget of the selective-SPN, and outputs a selective-SPN over $n$ variables with size $O(kn)$ (see Algorithm 1). The parameter $k$ can be chosen to balance the trade-off between expressiveness and size of the selective-SPN, where setting $k = 1$ gives a fully-factored distribution.

Algorithm 1 can be described as follows, proceeding layer-wise and keeping track of the number of partitions and the number of nodes per partition at each layer. At the leaf layer, we start with $n$ variable partitions and $c = 2$ nodes per partition; LITERAL returns deterministic leaf nodes 0 and 1 for each variable. At a product layer, CARTESIANPRODUCT creates $c^2$ product nodes for every pair of partitions, where $c$ is the number of nodes of a partition from the previous layer. As a result, the number of partitions are halved. At a sum layer, SUMNODE creates a sum node from an array of children nodes. We connect each child to one sum node, and set the number of sum nodes based on the size parameter $k$. We repeat sum and product layers until there is only one partition and one node. The total number of nodes is a function of the number of variables and the size parameter $k$.

**Theorem 2.** *Algorithm 1 constructs a selective-SPN of size $O(kn)$ in time $O(kn)$.*

The proof is in the Appendix and contains more details and explanation of the algorithm.

---

**Algorithm 1:** Constructing a Selective-SPN

---

**Input:** An integer $n$ (the number of variables), and an integer $k$ (the size budget for the SPN).
**Constraint:** Integer $n$ is a power of 2. Integer $k$ is an even power of 2.
**Output:** A selective-SPN over $n$ variables with size $O(kn)$.

```
 1  D, c ← {}, 2
 2  for i ← [0, n) do                                    /* leaf nodes */
 3  │    D[2i] ← LITERAL(−(i + 1))
 4  │    D[2i + 1] ← LITERAL(+(i + 1))
 5  while n > 1 do
 6  │    D' ← {}
 7  │    if c > k^{1/2} then                             /* sum nodes */
 8  │    │    r ← c/k^{1/2}
 9  │    │    for i ← [0, cn/r) do
10  │    │    │    D'[i] ← SUMNODE(D[ri : r(i + 1)])
11  │    │    c ← c/r
12  │    else                                           /* product nodes */
13  │    │    for i ← [0, n/2) do
14  │    │    │    d_1 ← D[2c(i + 0) : 2c(i + 1)]
15  │    │    │    d_2 ← D[2c(i + 1) : 2c(i + 2)]
16  │    │    │    D'[c²i : c²(i + 1)] ← CARTESIANPRODUCT(d_1, d_2)
17  │    │    c, n ← c², n/2
18  │    D ← D'
19  if c > 1 then  D[0] ← SUMNODE(D[0 : c])
Return: D[0]
```

---

## 5   Experiments

We experimentally validate the use of selective-SPNs in variational inference of discrete graphical models. We use Algorithm 1 to construct selective-SPNs, compute the exact ELBO gradient, and optimize the parameters of the selective-SPN (the edges under sum nodes) with gradient descent. Each ELBO computation takes $O(tm)$ operations. Since the selective-SPNs have size $O(kn)$, each optimization iteration takes $O(tkn)$ steps, where $t$ is the number of terms when expressing the log density of the graphical model as a polynomial, $k$ is a hyperparameter denoting the size budget, and $n$ is the number of binary variables. We present results on computing lower bounds of the log partition function of three sets of graphical models: Ising models, Latent Dirichlet Allocation, and factor graphs from the UAI Inference Competition. Experiments were run on a single GPU. Code can be found at `https://github.com/AndyShih12/SPN_Variational_Inference`.

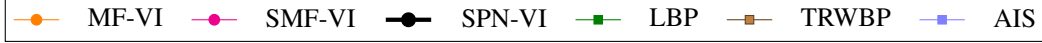

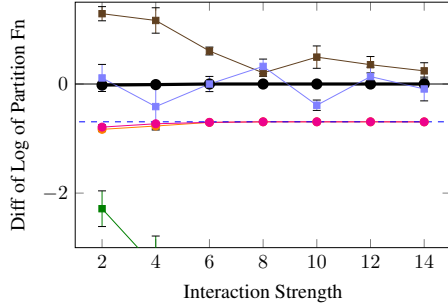

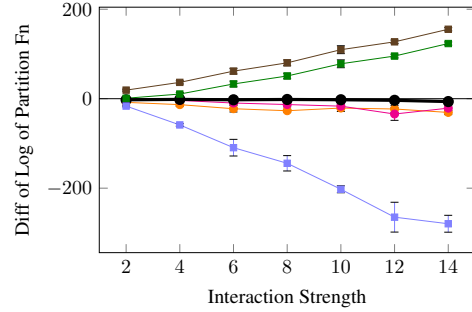

(a) $4 \times 4$ Ising model with positive interactions. We plot the difference from ground truth (black line at 0). MF-VI approaches $ln(1/2)$ (dotted line) since it only covers one of two symmetric modes for the positive interaction case.

(b) $8 \times 8$ Ising model with mixed interactions. We plot the difference from ground truth (line at 0).

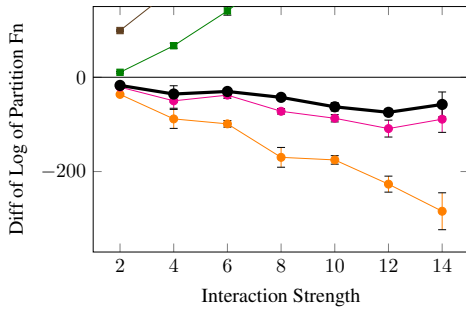

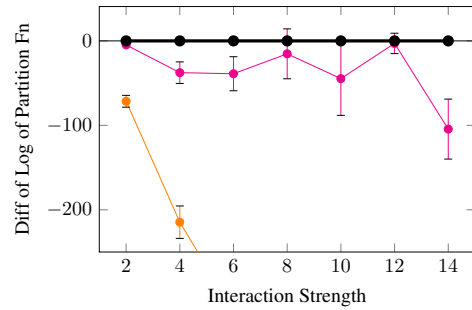

(c) $16 \times 16$ Ising model with mixed interactions. We plot the difference from ground truth (line at 0).

(d) $32 \times 32$ Ising model with mixed interactions. Ground truth is not computed, so we plot the difference from SPN-VI.

Figure 2: We compute the log partition function of Ising models of different sizes. We vary the interaction strengths along the x-axis. We plot the difference from ground truth for all grid sizes except for $32 \times 32$. Mean-Field (**MF-VI**), Structured Mean-Field (**SMF-VI**), and Sum-Product-Network (**SPN-VI**) are lower bounds, while Loopy Belief Propagation (**LBP**), Tree-Reweighted Belief Propagation (**TRWBP**), and Annealed Importance Sampling (**AIS**) are estimates. Points closer to 0 are better, and ones too far off are omitted.

## 5.1 Ising Models

We experiment on Ising models with randomly generated interaction strengths between the range $[-\gamma, \gamma]$, where $\gamma$ is a parameter that we vary. For each Ising model, we compare 6 different methods for approximating its partition function: variational inference with mean-field (MF-VI), structured mean-field (SMF-VI), selective-SPNs (SPN-VI), as well as Annealed Importance Sampling (AIS), Loopy Belief Propagation (LBP), and Tree-Reweighted Belief Propagation (TRWBP).

We experiment on Ising models of size up to $32 \times 32$ (up to 1024 variables and 1984 terms), and vary $\gamma$ from 2 to 14. For each size and each interaction strength $\gamma$, we report the estimated partition function for each of the methods, averaged over 4 randomly constructed Ising models. For the variational methods, we do multiple random restarts. Since each estimate is a lower bound, we report the maximum estimate from all the restarts. For AIS, we used 5 intermediate distributions [29]. For LBP and TRWBP, we used the implementations from libDAI [28]. We ran each method for 30 minutes, and plot the results in Figure 2. We can see that SPN-VI gives the best approximation of the ground truth for sizes $4 \times 4$, $8 \times 8$, and $16 \times 16$. We do not compute the ground truth for size $32 \times 32$, but we can see that SPN-VI gives the best lower bound compared to MF-VI and SMF-VI.

## 5.2 Latent Dirichlet Allocation

Next, we study Latent Dirichlet Allocation [1]. We assume that the prior parameters $\alpha$ and $\beta$ are given, and perform inference on the per-document topic distributions $\gamma$ and word topics $\phi$. Since the ELBO (cf. Eq. 15 in [1]) only has linear terms on $\phi$, we used a shallow mixture model with 16 selective mixtures – a special case of selective-SPNs with only one sum node at the root – as the variational distribution. In Figure 3, we see the improvement of using a selective mixture model over mean-field variational inference for 100 documents.

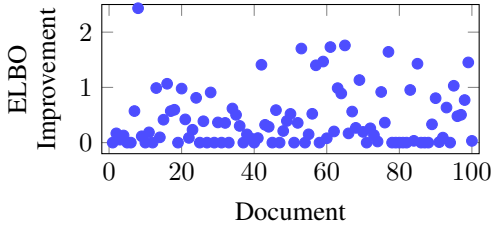

Figure 3: Improvement of ELBO on LDA when using selective mixtures instead of mean-field.

## 5.3 UAI Inference Competition

Lastly, we consider factor graphs from the 2014 UAI Inference Competition. We selected all the factor graphs from the competition with binary variables and no logical constraints. We compare the methods AIS, MF-VI, SMF-VI, SPN-VI, LBP, and TRWBP as shown in Table 1. The experimental setup is similar to that of Section 5.1. In general, SPN-VI is competitive with LBP, outperforming it on various DBN and Grid graphs. Additionally, SPN-VI gives a stronger lower bound than MF-VI and SMF-VI, while the other methods do not give a bound.

Table 1: Computing the log partition function of factor graphs from the 2014 UAI Inference Competition. The estimate closest to the ground truth is bolded, and the strongest lower bound is underlined.

| Graph | Vars/Terms | AIS | MF-VI | SMF-VI | SPN-VI | LBP | TRWBP | G. Truth |
|---|---|---|---|---|---|---|---|---|
| Alchm_11 | 440/4080 | 1219.01 | 1395.55 | 1395.55 | 1395.55 | **1395.98** | —[1] | 1396.01 |
| DBN_11 | 40/1680 | 103.87 | 134.10 | 134.10 | 134.10 | **134.66** | 319.33 | 134.77 |
| DBN_12 | 42/1848 | 107.69 | 144.26 | 142.11 | 144.85 | **145.32** | 339.14 | 145.40 |
| DBN_13 | 44/2024 | 114.55 | 149.55 | 146.43 | 151.22 | **152.25** | 406.20 | 153.25 |
| DBN_14 | 40/1680 | 168.96 | **348.05** | **348.05** | **348.05** | 348.05 | 348.06 | 348.04 |
| DBN_15 | 42/1848 | 167.37 | **351.40** | **351.40** | **351.40** | 351.40 | 352.15 | 351.41 |
| DBN_16 | 44/2024 | 178.83 | 378.45 | 378.45 | **382.48** | 378.45 | 383.22 | 382.48 |
| grid10x10 | 100/920 | 378.79 | 667.53 | 685.94 | **694.22** | 798.80 | 908.14 | 697.88 |
| Grids_11 | 100/1000 | 225.59 | 372.49 | 379.24 | **386.93** | 407.95 | 487.16 | 390.08 |
| Grids_12 | 100/920 | 404.30 | 664.22 | 681.54 | **691.34** | 798.80 | 908.14 | 697.88 |
| Grids_13 | 100/1000 | 446.38 | 735.89 | 742.79 | **763.40** | 804.64 | 965.38 | 767.50 |
| Grids_14 | 100/1000 | 555.44 | 1082.01 | 1114.93 | **1137.85** | 1200.23 | 1445.42 | 1146.14 |
| Grids_15 | 400/3840 | 392.83 | 632.14 | 639.52 | 657.99 | **677.15** | 800.23 | 671.74 |
| Grids_16 | 400/3840 | 574.74 | 1452.33 | 1457.20 | 1471.14 | **1551.70** | 1886.85 | 1531.49 |
| Grids_17 | 400/3840 | 810.67 | 2819.31 | 2830.10 | 2873.73 | **3112.30** | 3745.26 | 3020.95 |
| Grids_18 | 400/3840 | 1175.98 | 4199.07 | 4238.43 | **4304.62** | 4768.06 | 5610.62 | 4519.93 |
| relat_1 | 250/62500 | 720.68 | 735.10 | 735.10 | 735.10 | **735.20** | 746.14 | 735.21 |
| Seg_11 | 228/2924 | -318.63 | -63.45 | -63.24 | -61.46 | **-60.50** | -44.94 | -55.25 |
| Seg_12 | 229/2946 | -318.00 | -23.70 | -23.70 | -23.70 | **-23.69** | -23.57 | -23.69 |
| Seg_13 | 235/3058 | -319.24 | -78.42 | **-78.20** | -79.65 | -79.12 | -67.38 | -76.83 |
| Seg_14 | 226/2928 | -313.04 | -105.38 | -104.55 | -92.80 | **-91.32** | -83.40 | -90.94 |
| Seg_15 | 232/2988 | -366.68 | -74.90 | -74.75 | -73.72 | -72.75 | **-58.44** | -60.43 |
| Seg_16 | 231/2994 | -296.69 | -91.91 | -116.27 | -90.01 | **-88.68** | -77.39 | -87.79 |

# 6    Related Work

There is a large body of work on estimating variational objectives for discrete settings. Most existing approaches are Monte Carlo methods [27, 26, 13, 41, 42], typically relying on continuous relaxations, which introduce bias, or score function estimators [45], which have high variance. Also of interest are neural variational inference approaches [22] that estimate an upper bound of the partition function, and transformer networks for estimating marginal likelihoods in discrete settings [46].

The approach of computing the variational lower bounds analytically has generally been restricted to mean-field or structured mean-field [37]. One line of work has studied the use of mean-field for exact ELBO computation of Bayesian neural networks [15, 10]. More relevant works also consider graphical models with polynomial log-density, but are still restricted to fully factored variational distributions [12]. Our result shows that computing the ELBO can be tractable for the more expressive variational family of selective-SPNs.

In probabilistic circuit literature, moments of pairs of PSDDs can be computed in quadratic time on the size of the circuits [16]. Their result may be used to extend our analysis to log-densities parameterized by probabilistic circuits. However, PSDDs are less succinct than selective-SPNs, and the quadratic complexity may not be practical inside an optimization loop.

Most relevant to our work is the approach in [25] of compiling selective-SPNs (also known as arithmetic circuits) for variational optimization. Our work differs from [25] in two ways. First, we forgo the costly step of learning the SPN structure, and instead construct the SPN structure procedurally without considering the target graphical model. Second, we provide a more efficient computation of the variational lower bound. In particular, our method computes the gradient of the entropy in $O(m)$ – an improvement over their method which takes $O(m^2)$ operations, where $m$ is the size of the SPN. Since the variational lower bound improves with SPN size (with $m$ up to $1e5$ in our experiments), the speedup in calculating the entropy gradients is important.

# 7    Conclusion

We study the problem of variational inference for discrete graphical models. While many Monte Carlo techniques have been proposed for estimating the ELBO, they can suffer from high bias or variance due to the inability to backpropagate through samples of discrete variables. As such, in this work we examine the less-explored alternative approach of computing the ELBO analytically.

We show that we can obtain exact ELBO calculations for graphical models with polynomial log-density by leveraging the properties of probabilistic circuit models. We demonstrate the use of selective-SPNs as the variational distribution, which leads to a tractable ELBO computation while being more expressive than mean-field or structured mean-field. To improve scalability and ease of use, we propose a linear time construction of selective-SPNs. Our experiments on computing a lower bound of the log partition function of various graphical models show significant improvement over the other variational methods, and is competitive with approximation techniques using belief propagation. These findings suggest that probabilistic circuits can be useful tools for inference in discrete graphical models due to their combination of tractability and expressivity.

## Broader Impact

Our contributions are broadly aimed at improving approximate inference in graphical models. Our research could be used to develop more scalable and more accurate inference methods for machine learning models in general. As seen in our experiments, this includes handling models for physics, word/topic analysis, or image segmentation. Scaling to even bigger models can open up even more potential applications.

## Acknowledgments

We thank Antonio Vergari and Daniel Lowd for helpful discussions and clarifications regarding [25]. Research supported by NSF (#1651565, #1522054, #1733686), ONR (N00014-19-1-2145), AFOSR (FA9550-19-1-0024), and FLI.

## Footnotes

[1]Non-pairwise factors are not supported by the TRWBP algorithm in libDAI.

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
