[Supplementary Material]

# A   Proof of Theorem 2

*Proof.* First, we prove that the constructed circuit is a selective-SPN by showing it is selective and decomposable. We maintain these invariants at each layer: 1) the array $D$ store $cn$ nodes where $c$ is the number of nodes per partition and $n$ is the number of partitions and 2) the $c$ nodes of each partition have disjoint support. The invariants hold trivially at the leaf layer with $c = 2$. At a product layer, we do a Cartesian product between the $c$ nodes of partition $2i$ and the $c$ nodes of partition $2i + 1$ for $i \in [0, n/2)$ (lines 13-17). Since the children of each partition have disjoint support, the Cartesian product of children from two partitions forms $c^2$ product nodes with disjoint support. At a sum layer, each sum node is a mixture of $r$ children and each child is assigned to only one sum node, so setting $c \leftarrow c/r$ maintains invariant 1 and 2 (lines 8-11). Therefore, each sum node is selective, each product node is decomposable, and the constructed circuit is a valid selective-SPN.

To finish, we show that the constructed selective-SPN has size $O(kn)$. At every sum layer, we set $r$ so that there are at most $k^{1/2}$ sum nodes per partition. At every product layer, we have $c^2 \leq k$ product nodes per partition (line 7). It can be seen that there cannot be more than 2 consecutive sum layers, and that at every product layer the number of partitions halves. The total number of nodes/edges is therefore $O(k(n + \frac{n}{2} + \ldots + 1)) = O(kn)$. Each node/edge can be constructed in constant time, so the time complexity also scales as $O(kn)$. $\qquad\square$