[Reviews · NeurIPS 2020]

Review 1

Summary and Contributions: The authors present a new Variational Inference approach to computing the log partition of graphical models, using selective Sum-Product networks as tractable approximate model class. The proposed approach is compared against typical variational inference approaches (mean filed, structured prediction) on benchmarks instances, displaying state-of-the-art performance.

Strengths: The proposed approach is simple (hence easily reproducible), shows an interesting use of probabilistic circuits (as an approximation class) and leads to state-of-the-art performance. The proposed approach provides upper bounds (which most methods do not).

Weaknesses: The proofs of the results are left out (they appear in sup. material), some of which I could not follow. The experiments are not thorough; the leave out some important techniques (tree-reweighed bp, the combined approach implemented in libdai for the UAI 2010 competition).

Correctness: The theorem that shows that the variational bound is correct. I could not follow the proof of the second Theorem, which shows how to generate the structure. In particular I was not able to understand how the proof shows that the generated structured is selective. Yet, I believe this to be of less importance, as generating a selective structure is trivial if one follows a decomposition-like approach. The experiments are carried out correctly, but as mentioned above, could include more competitors to provide a more clear picture.

Clarity: The paper is well written.

Relation to Prior Work: The paper is well situated.

Reproducibility: Yes

Additional Feedback: Def 1 is not quite right: it is not the supports that are disjoint, but the scopes (either the supports are incomparable, as they are in different dimensions, or, if one considers their extension, they are not disjoint, as one is constant wr.t. to the missing variables) (page 4, line 151) Why is the work limited to binary variables? I could not find a need for this. (page 4, line 164) There is some ambiguity in the literature about a monomial being a product of variables (no constant), or a single-term polynomial. Better formalized it as this is rather important for the later development. In Lemma the citation is not in the proper format ([O'Donnel 2014] should be [27]) While the representation as monomials is common for pseudo-boolean functions is less common for graphical models in general; since there is not much difference, it would be better to assume a representation in terms of factors, which is more usual (and which is the case of the models used in the experiments). There are known results about computing moments and KL divergence in structured decomposable probabilistic circuits. Since entropy is KL div for two circuits with the exact same structure, these results apply in part to prove Theorem 1. This could be at least commented. While I understand that being efficiently computed as basic operations (sum, products, logs) implies being efficiently differentiable (by automatic diff software), it would be more interesting to prove Theorem 1 directly for the derivatives (since this is what is actually needed). In sec 5.2, I do not understand what are shallow mixtures with 16 selective mixtures? What is a selective mixture?


Review 2

Summary and Contributions: The authors present a variational distribution that allows exact ELBO computation on binary graphical models. This distribution is based on probabilistic circuits, more precisely, on Selective Sum Product Networks (SSPN). The properties of this SSPN allow them to compute expectations and entropy tractably. The authors then provide a learning algorithm and empirical evidence showing that SSPNs are competitive.

Strengths: The paper tackles an important problem in graphical model optimization. The

Weaknesses: The authors provide a model and evidence for binary data which indeed is discrete, yet the claim would lead a reader to believe this paper is about the general discrete setting. The methods presented might be extended to the discrete case, but empirical evidence is necessary and could be a nice follow-up contribution.

Correctness: The theoretical results are sound, and the evaluation is reasonable. Furthermore the combination of SPNs and the deterministic restriction fit well for the reduction in inference time and is a clever contribution.

Clarity: The paper is well written and clearly motivated.

Relation to Prior Work: The relation to prior work is reasonable, considering that exact computation of the ELBO is very rare.

Reproducibility: Yes

Additional Feedback: I have one question regarding the experiments from the UAI challenge in table 1, Why did you not compare to SMF-VI? It seems to me that this would be the closest in spirit to the SSPN approach. After reading the rebuttal, I still see the value of the paper even as a more applied improvement approach. Of course, improving the theoretical aspects would make it stronger.


Review 3

Summary and Contributions: In their paper, the authors propose a novel method for computing Evidence Lower Bound (ELBO) for a subclass of graphical models using selective (deterministic) SPNs.

Strengths: - the claims in the paper are well supported - using selective-SPNs for computing ELBOs is an interesting idea - the empirical evaluation considers various sources of graphical models - the contribution seems novel

Weaknesses: W1 The authors talk about the "full expressiveness" of probabilistic circuits but then proceed to fit a fixed probabilistic circuit (of limited expressiveness). Unless I missed something, while the inference is exact, the probabilistic circuit is (usually) an approximation. W2 Due to my limited background in variational inference, I miss a more detailed explanation of the steps in the inference pipeline after the construction of the SPN.

Correctness: Yes. Adding PSDDs as a competing method would be interesting but might be out of scope.

Clarity: Generally yes, but see (W2) an overview of the inference pipeline would be useful.

Relation to Prior Work: For the prior work mentioned in the paper the differences are clear, however, I wonder what the connection is to standard WMC based inference on circuits?

Reproducibility: Yes

Additional Feedback: Post rebuttal: Full expressiveness was addressed, pipeline more or less, WMC: I wish the authors could address this in a bit more detail but this might not be in scope for a rebuttal. Because the answers do not show major new deficiencies or advantages my score remains the same.


Review 4

Summary and Contributions: Post-rebuttal "For models with hidden variables, our method should be able to handle \sum_z p(x, z) as long as p(x, z) has the right structure for every x." -> it is a very strong assumption. I believe SMF should be in comparison, regardless of GPU implementation efficiency. It actually answers the question of how much structure we need from MF to SSPN. The only difference in the efficiency of computing the gradient of ELBO comparing to Lowd and Domingos (2010) is in computing the gradient of entropy. Computing entropy in one pass is the by-product of using selective-SPN vs. general ACs, not any particular novelty in the proposed method. ACs similarly use backpropagation to compute the expectation. =============== This paper proposes using probabilistic circuits, specifically selective SPNs, as a variational distribution for approximate inference. Similar to mean-field or structured mean-field approach, the suggested method results in a lower bound (tighter than the naive mean-field) of the partition function (hence an upper bound on the probabilities, which is less desirable). Using selective SPNs as a variational distribution is interesting in the sense that the lower bound on the partition function can be computed exactly. The alternative is sampling-based methods that have high variance (the paper also mentioned high bias, which is not often the case). The main limitation of the proposed method is that the exact computation of lower bound is limited to the models without hidden variables, which is less interesting for the community. Using probabilistic circuits as a variational distribution has been studied before by Lowd and Domingos (2010) [Approximate Inference by Compilation to Arithmetic Circuits, 2010]. Selective-SPNs share the same properties in computing the exact lower bound on partition function as arithmetic circuits, so I won't consider calculating the exact bound for selective-SPNs as a contribution of this paper. However, the main limitation of Lowd and Domingos (2010) is learning the structure of arithmetic circuits, which are done using sampling from the graphical model or pruning. The current paper proposes learning the structure randomly based on RAT-SPN, which would have more structure and for sure tighter bound than mean-field. In term of having bound, there are other methods that create upper bound on partition function and lower bound on probabilities (which are more desirable) such as classic tree-reweighted belief propagation (Wainwright et al.) or modern neural variational inference (Kuleshov and Ermon, 2017) which have not discussed in the paper. Inference network discussed in (Wiseman and Kim, Amortized Bethe Free Energy Minimization for Learning MRFs, 2019) is also sample free, although it does not provide any bound.

Strengths: Please see the summary.

Weaknesses: Please see the summary.

Correctness: Please see the summary.

Clarity: The paper is clear and well written.

Relation to Prior Work: Please see the summary.

Reproducibility: Yes

Additional Feedback: Modifying the contribution of the paper and addressing the mentioned relevant works. For Section 5.1, comparing to Kuleshov and Ermon (2017) would be very informative regarding the claims about the problems of sample-based methods. SMF-VI should also be in Table 1.

[Author Response · NeurIPS 2020]

1 Thank you all for the helpful reviews. We appreciate the acknowledgment that our paper is "well written and clearly
2 motivated" and "tackles an important problem in graphical model optimization". We're happy to hear that we have shown
3 an "interesting idea" and "clever contribution" that achieves "state-of-the-art performance" and is "easily reproducible".

4 We start by addressing Reviewer 4's concern that "calculating the exact bound for selective-SPNs" is not a "contribution
5 of this paper". Thank you for pointing out the relevant work by Lowd and Domingos (2010). Our problem setup is
6 indeed similar. However, our method of computing the ELBO gradient is in fact **novel** and is much **more efficient** than
7 their method, on two fronts.

    1. First, using their notation, their computation of all gradients takes $O((n+m)e)$, whereas our computation of
       all gradients takes $O(ne)$, since $n \rightarrow t$ and $e \rightarrow kn$ in our notation (line 226 in our paper). Since they use
       small circuits, they claim $m < n$ and $O((n+m)e) = O(ne)$. However, circuits in recent years have grown to
       hundreds of thousands in size (and in our experiments $m \gg n$), so $O(ne)$ **is much better than** $O(me)$.

    2. Second, and equally important, is that we compute all the gradients in one pass of the circuit thanks to
       backpropagation, which allows for **GPU optimization**. For large models, we saw a 5-10x speedup with GPUs.
       Their method computes the gradient of each individual parameter separately (cf. Eq5 in their paper) and
       requires fixing different parts of the circuit constant, which cannot be easily optimized with GPUs.

16 As such, our method is much faster both in theory and in practice, and is one of the main contributions of this paper.

17 **TRWBP:** Thanks for the suggestion of comparing against tree-reweighted BP. We use TRWBP from libDAI, using
18 10000 random spanning tree samples. For Fig2, TRWBP does better than SPN-VI for 4x4 but worse for 8x8, 16x16,
19 and 32x32. For Tab1, TRWBP is generally worse than LBP. For example (will include everything in final version):

| DBN11 | DBN13 | DBN15 | grid10 | Grids11 | Grids13 | Grids15 | relat. | Seg11 | Seg13 | Seg15 |
|---|---|---|---|---|---|---|---|---|---|---|
| 319.33 | 406.20 | 352.15 | 908.14 | 487.16 | 965.39 | 800.23 | 746.14 | -44.94 | -67.38 | -58.44 |

21 **SMF-VI:** We could not find a competitive implementation of SMF-VI that uses GPUs. We had to implement our own
22 version of SMF-VI that is GPU-optimized (see the supplementary zip file), but it only works for degree-2 interactions
23 (as shown in Fig2 for Ising models). We were not able to easily extend our GPU-optimized impl. to general graphical
24 models for Tab1, and comparing against a non-GPU impl. seems unfair (but happy to include if reviewers want).

25 **R1** "leave out tree reweighted BP": Thanks, please see **TRWBP**.
26 "proofs of the results", "generated structured is selective": Ok, we will clarify this. Yes, the main intuition is to generate
27 partitions of the support using partitions of the children variables, in a decomposition-like approach.
28 "Def 1" disjoint scopes: Good point, we will reword this to say scope.
29 "limited to binary": The circuits can be over categorical variables as well. We focused on binary variables for simplicity.
30 "(page 4, line 164) monomial": Ok, we will clarify that there is a constant in our setting.
31 "citation is not in the proper format": Thanks, we will fix the formatting.
32 "better to assume a representation in terms of factors": Ok we will keep this consistent.
33 "moments in structured decomposable probabilistic circuits": Yes, we did comment on this at the end of Related Work.
34 "prove Theorem 1 directly for the derivatives": Good point, we will consider this.
35 "What is a selective mixture?": This refers to shallow SPNs of depth 2.

36 **R2** "general discrete": Sorry, will clarify that our framework can be extended to general discrete settings by using
37 discrete leaf distributions, but our experiments currently are only binary.
38 "Why did you not compare to SMF-VI": Please see **SMF-VI**.

39 **R3** "full expressiveness": Yes, you understood correctly. We will clarify that the family of models is fully expressive
40 but the circuit we construct is an approximation.
41 "detailed explanation...inference pipeline": Ok we can include more detail. Briefly: we backprop on the exact ELBO to
42 get gradients w.r.t. SPN params, and perform gradient steps to optimize the lower bound estimate of partition function.
43 "WMC based inference": Should be possible to convert WMC to an equivalent PGM problem and apply our method.

44 **R4** "limited to the models without hidden variables": For models with hidden variables, our method should be able to
45 handle $\sum_z p(x,z)$ as long as $p(x,z)$ has the right structure for every $x$.
46 "studied before by Lowd and Domingos": Thanks for the reference. Please see above (ours is much more efficient).
47 "tree-reweighted belief propagation": Please see **TRWBP**.
48 "neural variational inference": Thanks for the reference. Unfortunately their repository did not include their code for
49 Ising models. Nevertheless, their paper clarifies that the upper bound "will not be directly applicable to highly peaked
50 and multimodal distributions...such as an Ising model". So in their experiments they only bound the partition function
51 of an **approximation** of the original Ising model, and only scale to size 5x5 (we scale to 32x32).
52 "Inference network discussed in Wiseman...": Thanks again! We will discuss this too in the related work.
53 "SMF-VI should also be in Table 1": Please see **SMF-VI**.

[Meta-Review · NeurIPS 2020]

The work develops a variational approximation for the log partition function of binary-variable graphical models using selective Sum-Product networks. The theoretical foundations rest heavily on Lowd and Domingos (2010) but the application (using S-SPN's as variational aproximants) is novel and the practical results look very promising. The work received mixed reviews but the overall consensus was positive. The reviewers agree that the method is novel and that the ability to produce a bound on the partition function (albeit a lower one) is useful. The main weakness is the lack of an experimental comparison with competing post-mean-field-and-loopy-BP methods such as Tree-Reweighted BP, neural variational inference and the Wiseman reference. At a minimum the final paper needs to discuss these references and include the TRWBP experiments from the rebuttal and (non-GPU if necessary) SMF results in table 1.